# Development of Dynamic Model for Real-Time Monitoring of Ripening Changes of Kimchi during Distribution

**DOI:** 10.3390/foods9081075

**Published:** 2020-08-07

**Authors:** Ji-Young Kim, Byeong-Sam Kim, Jong-Hoon Kim, Seung-Il Oh, Junemo Koo

**Affiliations:** 1Research Group of Consumer Safety, Korea Food Research Institute, 245, Nongsaengmyeong-ro, Iseo-myeon, Wanju-gun, Jeollabuk-do 55365, Korea; jykim@kfri.re.kr (J.-Y.K.); bskim@kfri.re.kr (B.-S.K.); jhkim@kfri.re.kr (J.-H.K.); dr51@kfri.re.kr (S.-I.O.); 2Department of Mechanical Engineering, Kyung Hee University, 1732, Deogyeong-daero, Giheung-gu, Yongin-si, Gyeonggi-do 17104, Korea

**Keywords:** kimchi ripening, acidity, predictive model, dynamic model, mean kinetic temperature

## Abstract

This study describes the development of a method for predicting the ripening of Kimchi according to temperature to provide information on how the ripening of Kimchi changes during distribution. Various Kimchi quality factors were assessed according to temperature and time. The acidity (lactic acid %) was selected as a good freshness index, as it is dependent on temperature and correlates strongly with the sensory quality evaluation. Moreover, it is easy to measure and reproducible in the field. The maximum value of acidity in the stationary phase was observed to increase with the storage temperature. A predictive model was developed using the Baranyi and Roberts and Polynomial models to mathematically predict the acidity. A method using the mean kinetic temperature (MKT) was proposed. The accuracy of the model using the MKT was high. It was confirmed that there is no great variation in the maximum acidity, as MKT does not change much if the temperature changes in the stationary phase where the maximum acidity is constant. This study provides important information about the development of models to predict changes in food quality index under fluctuating temperature environments. The developed kinetic model uniquely treated the quality index at the stationary phase as a function of MKT. The predictions using the food temperature histories could help suppliers and consumers make a reasonable decision on the sales, storage, and consumption of foods. The developed model could be applied to other products such as beef for which the quality index at the stationary phase also changes with temperature histories.

## 1. Introduction

Kimchi is a traditional Korean food consisting of fermented salted vegetables (i.e., cabbage, radish) and ingredients such as spring onion, garlic, ginger, and red pepper powder (CODEX STAN 223-2001). Other fermented vegetable products similar to Kimchi are sauerkraut (fermented cabbage), pickled cucumbers, and Tsukenmono (preserved vegetables from Japan). Notably, fermented vegetable products are known for various health benefits around the world, and many studies have been conducted to prove such claims [1,2,3].

While Kimchi was traditionally made in the home, it is now commodified and mass-produced owing to the rise of the nuclear families eating out and group meals. Korean cabbage Kimchi accounts for 75% of domestic Kimchi sales, from Korean Won (KRW) 677.4 billion in 2014 to KRW 807.3 billion in 2016 and have been increasing yearly. In addition, Kimchi exports reached US$ 97.45 million in 2018, a 20% increase compared to the year before with the number of export destinations reaching 68 countries [4].

Since Kimchi is a non-sterilized product, its fermentation continues during distribution. However, the taste and edibility period depend on the degree of fermentation. This makes it very difficult to ensure the quality of the product. Food is inevitably exposed to various temperature conditions from production to delivery. The temperature of the workshop where Kimchi is manufactured is kept below 10 °C, and the storage warehouse where it is kept until dispatch is typically 0–2 °C. Domestic sales are shipped in a refrigerated truck (0–5 °C) and then kept in refrigerated retail displays (2–10 °C) while exported Kimchi is transported by ship in a temperature controlled container. Extreme temperature control is required as the product is exposed to the external environment upon quarantine, customs clearance, warehouse transfer, and sale upon arrival.

Owing to the diverse temperature changes during distribution, the maturity and edibility of Kimchi cannot be confirmed until the packaging is opened. To solve this problem, Jung et al. [5] developed a chitosan-based carbon dioxide indicator to evaluate the extent of ripening of Kimchi according to the concentration of carbon dioxide inside the packaging. Kim et al. [6] aimed to identify the fermentation stage of Kimchi by evaluating the volatile organic compounds (VOCs) released using a colorimetric indicator and correlating the results with the physicochemical state of the Kimchi. These methods allow the consumer to roughly check the ripening of Kimchi via an indicator in the packaging at the time of purchase, but they do not allow quantitative information about the detailed ripening status. Therefore, it is important to inform consumers about the quality of their Kimchi by predicting how the freshness is affected by exposure to different temperatures and how long a certain level of quality can be maintained under a specified storage temperature.

The effects of external factors on the physical, chemical, and microbiological changes within food can be mathematically predicted and used for food safety and quality control assessment [7,8,9]. Predictive models are developed using a combination of primary and secondary models. The primary model predicts the change in food quality index over time, and the secondary model considers the temperature dependence of the parameters calculated in the primary model. Subsequently, the accuracy of the model is verified by comparing it with observational data that was not used to develop the model (either fixed-or fluctuating-temperature quality observations) [10,11]. Such predictive models have been actively studied to surmise the growth or decline of microorganisms on food exposed to dynamic temperature changes [12,13,14,15]. However, these models could only describe the quality index change for the cases with a constant value at the stationary stage. They cannot be used for the cases where the quality index values at the stationary phase vary with storage temperature such as Kimchi acidity. Similar phenomena have been reported with microbial growth in beef [16,17]. Jaisan and Lee [18] developed a kinetic model of Kimchi acidity change for the confining range between the lag and exponential growth phases, neglecting the stationary phase to avoid the difficulty of modeling it. Thus, for the case of the stationary phase, it is necessary to develop a methodology for producing a model that can describe acidity change from the lag to the stationary phases with the quality index varying with temperature history.

In this study, we analyzed the change of Kimchi quality according to temperature throughout the distribution process with several quality indices. Subsequently, we selected acidity as an appropriate freshness index that is dependent on temperature and has a high correlation coefficient with the sensory quality. Moreover, the selected freshness index is easily measured and reproducible in the field. We developed a dynamic prediction model to identify the freshness of Kimchi in a fluctuating-temperature environment (such as during distribution) by predicting mathematically selected quality index changes. In addition, we proposed a method for constructing food quality models using the mean kinetic temperature (MKT). The MKT represents the temperature history of the food at a given time as a single value. Finally, we verified the validity of the model by comparing the model prediction results with actual acidity measurements obtained when the Kimchi was exposed in a real fluctuating temperature environment.

The developed kinetic model uniquely treated the quality index at the stationary phase to be a function of MKT. It could successfully reproduce the observations under both constant and varying temperature conditions. This will help the suppliers decide on the sales and disposal of Kimchi in circulation based on the model prediction using the monitored temperature histories for their products. Both suppliers and consumers could prepare the temperature histories to make Kimchi of their favorite flavor. The developed model can also be applied to other products such as beef for which the quality index at the stationary phase changes with temperature histories.

## 2. Materials and methods

### 2.1. Sample Preparation

Kimchi samples used in this experiment were packed in 500 g units in polyethylene (PE) films immediately after production in a Kimchi manufacturing company (D Company, Seoul, Korea) and were transported in a refrigerated truck (2–5 °C). On arrival, the samples were stored in refrigeration units (Rei-technology Corporation, Anyang, Korea) at 0, 5, 10, and 20 °C. To reflect seasonal changes, Kimchi was purchased and tested in spring (March to May), summer (August to September), and winter (November to January), and the tests were labeled as experiments 1, 2 and 3, respectively.

According to the Codex Alimentarius, a collection of international food standards, the composition of acidity for fermented kimchi should not be more than 1%. Accordingly, the end of storage for Kimchi was generally selected to be the time for the acidity to approach the value of 1%. However, the observation period was extended to consider the plateaus of the stationary phase in this study. Kimchi generates carbon dioxide in the process of fermentation, and the polyethylene-made wrap could burst. These were considered in choosing the end of storage.

### 2.2. Quality Characteristics for Kimchi Ripening Index

To develop a prediction model that scientifically identifies the ripening of Kimchi during distribution, we first conducted experiments to evaluate a suitable ripening index. The experimental parameters chosen to reflect characteristics of Kimchi were pH, total acidity, Hunter color, hardness, aerobic count plate, lactic acid bacteria, and sensory characteristics.

#### 2.2.1. PH and Acidity

A 500 g unit of Kimchi was pulverized in a blender for 1 min and then filtered with gauze. The pH of the sample was then measured using a pH meter (TA-70, DKK-TOA Corporation, Tokyo, Japan). The acidity was determined by measuring the amount of 0.1 N (*w*/*v*) NaOH consumed to neutralize the pH to 8.2 after taking 20 mL of Kimchi filtrate. This value was then converted into lactic acid content (%, *w*/*w*) using Equation (1).
(1)$$\mathrm{Titratable}\;\mathrm{acid}\;\left(\%\right)=\frac{0.009\;\times \;0.1\;N\;\mathrm{NaOH}\;\left(\mathrm{mL}\right)\;\times \;0.1\;N\;\mathrm{NaOH}\;\mathrm{factor}}{\mathrm{Sample}\;\mathrm{weight}\;(g)}\times 100$$

#### 2.2.2. Hunter Color

The chromaticity of the Kimchi was measured using a color meter (CR-200, Minolta Co., Osaka, Japan). A 100-g sample of Kimchi was pulverized for 1 min in a blender then placed in a petri dish (35 × 20 mm) for the measurement. The color values were expressed as *L* (lightness), *a* (redness), and *b* (yellowness) after adjusting to standard plate (*L* = 97.75, *a* = 0.49, *b* = 1.96) before each measurement.

#### 2.2.3. Texture

The texture of the Kimchi was evaluated by measuring the hardness using a texture analyzer (TA-XT1, Stable Micro System Ltd., Godalming, England), and analyzed by the puncture test. Cabbage samples of 3 × 3 cm dimensions were cut from the area 5 cm below the root of the third cabbage leaf. For the measurement, the maximum strength received while penetrating 100% from the central portion of the crust was measured. The test conditions were as follows: a 3-mm cylindrical probe was used; the pre-test speed, test speed, and post-test speed were set to 3.0, 3.0, and 10.0 mm/s; and the distance was 15 mm.

#### 2.2.4. Microbiological Analysis

To assess the aerobic and lactic acid bacteria, 10 g of Kimchi leaf and stem were taken and mixed with 90 mL of sterilized 0.85% NaCl solution. The mixture was homogenized in a bag mixer (Interscience Inc., Saint-Nom-la-Bretèche, France) for 1 min. One mL of the sample was taken and diluted stepwise with 9 mL of sterile 0.85% NaCl solution. For the aerobic count plate, 1 mL of the diluted solution was inoculated in a 3M Petrifilm plate (3M Co., Saint Paul, MN, USA) and cultured at 37 °C for 48 h, then the number of viable aerobic bacteria were measured in Colony-forming units (CFU/g). The number of lactic acid bacteria were also measured in Colony forming units (CFU/g) after culturing at 35 ± 1 °C for 48 h in MRS broth (Difco, Detroit, MI, USA).

#### 2.2.5. Sensory Analysis

The sensory evaluation of Kimchi was carried out using a 9-point scale for 20 trained sensual assessors. In this case, the trained participants are selected from staff and researchers (5 males, 15 females, aged 30–60) at the Korea Food Research Institute. Beforehand, the participants were trained to differentiate kimchi qualities based on temperature and evaluate the appearance (softness), sour smell, sour taste, crunchiness, and overall ripeness of the Kimchi. For instance, “appearance” pertains to the color shade-softness of the Kimchi where “sour smell” described the sensation of typical generated flavor. The taste was used to obtain desired sour effects, and the ripeness was used to draw an overall freshness of the product. The intensity of each item was ranked from 1 (very low) to 9 (very strong); 3, 5, and 7 points were awarded for low, normal, and high intensities, respectively. The 3-g samples were provided in white polyethylene cups and numbered with three-digit randomly. Each sample was served one by one at room temperature.

#### 2.2.6. Statistical Analysis

The results of the sensory evaluation were assessed using variance analysis conducted on SPSS Statistics 20 (IBM, Armonk, NY, USA). The significance of the difference was verified using Duncan’s multiple range test at the *p* < 0.05 level. In addition, Pearson’s correlation coefficient was used to correlate the quality characteristics of Kimchi and sensory ripening evaluation indices.

### 2.3. Model Development

#### 2.3.1. Primary Model

Acidity was selected as the most suitable ripening index of Kimchi during storage. The acidity shows sigmoidal growth with time. The Gompertz, Hill and Wright, and Logistic, and Baranyi and Roberts models can be used to express sigmoidal growth. Herein, the Baranyi and Roberts model was selected as the primary model to express this change.
(2)$$\frac{dN}{dt}=\frac{Q\left(t\right)}{1+Q\left(t\right)}\cdot \mu_{\max}\left(T\right)\cdot \left[1-\frac{N}{N_{\max}}\right]\cdot N$$
(3)$$\frac{dQ}{dt}=\mu_{\max}\left(T\right)\cdot Q\left(t\right)$$
where N is the acidity (%); Q is a factor indicating the physiological food quality; t is elapsed time; T is food temperature; $\mu_{\max}$ is the maximum acidity growth rate at temperature T; and $N_{\max}$ is the maximum acidity value at temperature T (%). The initial conditions of $N\left(t=0\right)=N_{0}$ and $Q\left(t=0\right)=Q_{0}$ were used to integrate the first-order differential equations in Equations (2) and (3). Under the fixed temperature condition, the maximum acidity value at a given temperature condition ($N_{\max}$) does not change.

#### 2.3.2. Secondary Model

The secondary model describes the effect of environmental conditions on the parameters analyzed in the primary model. The most commonly used models are the Polynomial, Square root, and Ratkowski models. Herein, the Polynomial model was used to analyze the influence of $\mu_{\max}$ on temperature.
(4)$$\mu_{\max}\left(T\right)=a_{0}+a_{1}T+a_{2}T^{2}$$
where $a_{0}$, $a_{1}$, and $a_{2}$ are variables for minimizing the difference between the experimental and predicted values, determined using the optimization packages [19,20] of the open-source statistical program, R [21].

#### 2.3.3. Dynamic Model Using Mean Kinetic Temperature (MKT) in Fluctuating-Temperature Environments

A dynamic model was developed to predict the change in acidity (%) under fluctuating temperature conditions. First, the acidity values were measured under two fluctuating temperature profiles. The first fluctuation was between 0 and 10 °C at 24 h intervals for 20 days, and the second was between 5 and 15 °C at 24 h intervals for 14 days.

If $N_{\max}$ is constant, it is easy to predict the change in acidity under fluctuating temperature conditions, as the changes of N and Q over time can be obtained considering the influence of the instantaneous temperature on $\mu_{\max},\;$as described by Corradini et al. [22]. However, if $N_{\max}$ is a function of temperature, then $N_{\max}$ must change according to the change in temperature. That is, $N_{\max}$ should decrease with time as the temperature decreases in the stationary phase where N is constant. However, the $N_{\max}$ value observed in a given phase does not decrease even if the temperature decreases. Therefore, we devised a dynamic model which is modified to $N_{\max}=N_{\max}\left(T_{\mathrm{MKT}}\right)\;$by using the MKT. The MKT is obtained by integrating the effects of various temperature changes over a certain period on the acidity value; therefore, it tends not to change easily when the temperature changes instantaneously. Thus, $N_{\max}$ does not change according to the instantaneous temperature, as it is determined by the overall temperature history, as shown in Equation (5).
(5)$$N\left(t\right)-N_{0}=\int_{t=0}^{t}\frac{dN}{dt}dt=\int_{t=0}^{t}\frac{Q\left(t\right)}{1+Q\left(t\right)}\cdot \mu_{\max}\left(T_{\mathrm{MKT}}\right)\cdot \left[1-\frac{N}{N_{\max}\left(T_{\mathrm{MKT}}\right)}\right]\cdot Ndt$$

Herein, we calculated MKT from the acidity value at time t after identifying the relationship between T and N over time under fixed temperature conditions (0, 5, 10, 15, and 20 °C). The MKT at the starting point of the fluctuating temperature condition, $t_{0}$, is equal to the initial temperature under the fixed temperature condition. Therefore, the rate of acidity change at $t_{0}$ can be obtained using Equations (2) to (4) at the initial temperature and MKT. The acidity value $N_{x}$ at time $t_{x}$ can be obtained considering the elapsed time. For a given acidity value $N_{1}$ at time $t_{1}$, the constant temperature profile that best matches the fluctuating temperature profile is determined as the MKT. An example of the MKT calculation is shown in Figure 1. After five days, the acidity of the Kimchi was 0.546, and the MKT was 9.77 °C. Please refer to Kim et al. [23] for the detailed process of MKT calculation.

#### 2.3.4. Comparison between Observations and Predictions

The accuracy of the predicted values was verified by comparing them to the observed values using the bias factor $\left(B_{f}\right)$ and accuracy factor $\left(A_{f}\right)$ (Equation (6)).$\;B_{f}$ and $A_{f}$ are equal to 1 when the predicted and observed values are exactly the same; thus, the closer to 1, the higher the accuracy of the model [24].
(6)$$A_{f}=10^{\frac{\sum \left|\log\left(C_{\mathrm{predicted}}/C_{\mathrm{observed}}\right)\right|}{n}},\;B_{f}=10^{\frac{\sum \log\left(C_{\mathrm{predicted}}/C_{\mathrm{observed}}\right)}{n}}$$

## 3. Results

### 3.1. Changes in Quality Characteristics during Storage of Kimchi

#### 3.1.1. PH and Acidity

The pH of the Kimchi decreased over the fermentation period from the initial value of 5.93. However, the rate of decrease was faster as the storage temperature increased (Figure 2a). At 20 °C, the pH decreased rapidly to 4.43 in just two days and to 3.83 in ten days. At 10 and 5 °C, the pH decreased to 4.14–4.24 in 10 and 20 days, respectively, and then remained at 4.05–4.15 until the last day of storage. At 0 °C, the pH remained at 5.93–6.06 until the 14th day of storage and then decreased to 4.57 by the 28th day, after which it was maintained at 4.22–4.37 until the final day of storage (day 63). The decrease of pH during the fermentation process is due to the increase in the formation of various organic acids, including lactic acid, as the fermentation progresses. On the other hand, it was reported that the pH does not decrease continuously due to the buffering action of the free amino acids in Kimchi liquid [25]. This was also consistent with the results of this study.

The change in acidity showed a sigmoidal increase over time (Figure 2b). The Kimchi rapidly produces lactic acid and other organic acids from saccharides in the raw materials or spices. Subsequently, there is a lag phase in which the acidity did not easily increase due to moisture generation from the cabbage and the growth of aerobic microbes in the early stages of ripening. Finally, the acidity is maintained at a constant level. The acidity increased over the fermentation period at all temperatures, although the rate of increase was faster at higher temperatures. At 20 °C, it entered the exponential growth phase without showing lag. The amount of lactic acid increased sharply from 0.24% to 1.11% within 7 days of storage. Following this, the acidity was maintained at 1.2–1.3% until the tenth day. At 10 and 5 °C, the acidity entered the exponential growth phase after a certain lag phase and increased to 0.95–0.96% after 14 and 35 days of storage, respectively. Subsequently, the acidity remained between 0.9–1.1% until the last day of storage (day 30 and 50, respectively). At 0 °C, the acidity increased gradually without an exponential phase and remained at 0.81–0.85% from the 49th to the final day. These results were similar to those of other studies [5,26,27,28]; thus, the maximum acidity value of Kimchi is affected by temperature.

#### 3.1.2. Texture

Hardness tests were performed on the Kimchi as a measure of texture (Figure 2c). The initial hardness after fermentation was 14.7 N. There was a rapid decrease of hardness at higher fermentation temperatures. The hardness decreased to 7.8, 7.8, and 18.6 N after 8, 16, and 35 days at 20, 10, and 5 °C, respectively. The texture of Kimchi consists of pectin and fibers that make up the structure of Chinese cabbage. The tissue softens as the soluble pectin component increases during ripening [29]. Although hardness is a suitable descriptor for the freshness of Kimchi, it is difficult to measure objectively, as the type and structure of the cabbage, and therefore the hardness, differ depending on the position of the leaf in the cabbage when grown, as well as the variety, growing season, manufacturing method, and measurement area.

#### 3.1.3. Hunter Color

The color of Kimchi was assessed during fermentation. The L value, indicating lightness, showed little change over the course of the storage period. In contrast, the a value, indicating redness, showed a tendency to increase over the fermentation period; moreover, the rate at which it increased was faster at higher temperatures. When stored at 20 °C, a increased from 14.78 to 21.12 in just 4 days and then remained in the 20–21 region until the last day of fermentation. It increased to 20.08 by the 16th day of storage at 10 °C, to 20.35 by the 25th day at 5 °C, and to 20.02 by the 56th day at 0 °C. These results showed that the higher the fermentation temperature, the faster the change in redness is. The b value, indicating the yellowness of Kimchi, increased in the early stage of storage at all temperatures; however, there was no significant difference thereafter. The ANOVA analysis of the data revealed that the Hunter color values were not an adequate quality index to differentiate the food quality change over time with precision. The details of the analysis and graph related to Hunter color values were not included for the sake of presentation’s clearness.

#### 3.1.4. Microbiological Analysis

The microbial colonies in Kimchi were affected by the fermentation temperature, as shown in Figure 3a. The higher the temperature, the faster the growth rate and higher the maximum number of bacteria. In the aerobic count plate, when stored at 20 °C, the number of aerobic bacteria increased from the initial value of 5.02 log CFU/g to 8.52 log CFU/g by the third day of storage. Subsequently, the value gradually decreased until 7.85 log CFU/g by the final day of storage. At 10 °C, the maximum bacterial count reached 7.71 log CFU/g after six days of storage and then decreased gradually to 7.04 log CFU/g after 30 days. At 5 °C, the maximum number of bacteria reached 6.44 and 6.20 log CFU/g after 10 and 42 days, respectively. At 0 °C, the maximum bacterial count remained constant.

For the lactic acid bacteria (Figure 3b), the time it took to reach the maximum number of bacteria and the pattern of maintaining it differed depending on the storage temperature. Unlike the aerobic count plate, the maximum number of bacteria reached 7.5–8.0 log CFU/g even at low temperatures (0 and 5 °C). The main fermenting microorganism of Kimchi is the lactic acid bacteria *Leuconostoc* spp., which gives it its distinctive taste and smell. Since *Leuconostoc* spp. grows well even at low temperatures, it was the most-observed bacteria on all storage temperatures.

#### 3.1.5. Sensory Evaluation

Table 1 shows the results of variance analysis for the degree of softness, sour smell, sour taste, crunchiness, and overall ripening of Kimchi according to fermentation temperature. When stored at 20 °C, the intensity of softness, sour smell, and sour taste increased significantly until the 5th day of fermentation; at 10 °C, these indices increased until the 16th day and then either increased or decreased with showing no significant differences (*p* < 0.05). At 5 and 0 °C, the same indices showed significant increases until the 30th and 56th day of storage (*p* < 0.05). The crunchiness was significantly decreased until the 5th and 12th days of storage at 20 and 10 °C, although it showed no significant difference afterwards. At 5 and 0 °C, the crunchiness continued to decrease until the last day of storage (*p* < 0.05).

As a result of analyzing the overall ripening of Kimchi based on its appearance, taste, sourness, and texture, it was determined that the Kimchi stored at 0, 5, and 20 °C showed a significantly higher degree of ripening over the fermentation period (*p* < 0.05). At 10 °C, the degree of fermentation substantially increased until the 16th day; subsequently, there was no significant difference observed (*p* < 0.05). The variation of the degree of ripening over time is shown in Figure 4.

### 3.2. Selection of Ripening Index of Kimchi

We analyzed the correlation between the change of Kimchi quality and the sensory ripening evaluation index to select a quality index that can objectively determine the ripening of Kimchi during fermentation (Table 2). There were significant correlations (*p* < 0.01) between the pH and ripening of Kimchi at each storage temperature, with correlation coefficients of −0.920, −0.887, −0.884, and −0.862 at 0, 5, 10, and 20 °C, respectively. The correlation coefficients between acidity and ripening were 0.948, 0.932, 0.947, and 0.946 at 0, 5, 10, and 20 °C, respectively, again showing significant correlation at all temperatures (*p* < 0.01). However, the correlation was higher at higher temperatures, suggesting that the change in acidity is highly dependent on temperature and that acidity has a linear correlation with time in the region of maximum acidity increase. Previous studies [5,30,31] that identified ripening stages of Kimchi based on total acidity have reported that the acidity of non-fermented Kimchi increases to 0.2–0.4% in the early phase of fermentation, 0.5–0.8% in the moderately or optimally fermented stage, and over 1.0% when over-fermented. The acidity value measured in this study correlated well with the results of the sensory test, since the overall ripening value is 1–3 in the non-fermented stage with 0.2–0.4% acidity (low ripening stage), 4–6 in the moderate fermented stage with 0.5–0.8% acidity (moderate ripening stage), and 7+ in the over-fermented stage with over 1.0% acidity (high ripening stage). Therefore, acidity is an appropriate indicator for determining the Kimchi ripening status.

The correlations between chromaticity and overall ripening were analyzed according to storage temperature. The a values, indicating redness, had correlation coefficients of 0.807, 0.835, 0.536, and 0.603 at 0, 5, 10, and 20 °C, respectively, showing significant correlation (*p* < 0.01). The L and b values, indicating brightness and yellowness, had no significant correlation with overall ripening. Although the Kimchi color intensity tended to increase over the fermentation period, the sensitivity is dependent on ingredients such as red pepper powder and fish sauce, as well as the measured part of the Kimchi, (stem or leaf of cabbage), or Kimchi liquid. Therefore, the chromaticity is not an appropriate index for evaluating Kimchi ripening.

As a result of analyzing the correlation between microbiological changes and overall Kimchi ripening according to storage temperature, the number of aerobic bacteria in the aerobic count plate showed correlation coefficients of 0.892 (*p* < 0.01) and 0.595 (*p* < 0.05) at 0 and 10 °C, respectively, showing significant correlation, while no significance was observed at 5 or 20 °C. Lactic acid bacteria showed significant correlation at 0, 5, 10, and 20 °C with correlation coefficients of 0.860, 0.628, 0.674, and 0.665, respectively (*p* < 0.01, *p* < 0.05). Although lactic acid bacteria are the main microorganisms for the Kimchi fermentation, there are differences in the degree of bacterial growth depending on the fermentation temperature as there are various kinds of lactic acid bacteria that can grow [32]. Therefore, further research is needed before bacterial growth can be used as a representative index for Kimchi ripening.

### 3.3. Model Development

#### 3.3.1. Kinetic Model

A model was developed using acidity as a Kimchi ripening index. The acidity changes in a constant temperature environment show a sigmoidal shape with time. The acidity tends to increase over the fermentation period at all temperatures; the higher the temperature, the faster the rate of increase, and the higher the maximum acidity (Figure 5). The symbols in Figure 5 represent the measured data, and the lines show model fittings.

The acidity measurement results obtained in constant temperature experiments were corresponded by temperature to obtain fitted model constants and their confidence intervals. The results are shown in Table 3. The actual fitting process was performed by determining the constants that minimized the sum of squares of the differences between the data from experiments 1 to 3 and the fitted values of the model. The DEoptim [19] and nls.lm [20] packages (an optimization tool of R [21], open statistical software) was used for this purpose. From the regression analysis of the Baranyi and Roberts model, the maximum acidity $N_{\max}$ increases linearly with temperature; its regression equation is shown in Equation (7).
(7)$$N_{\max}=N_{\max\_\mathrm{intercept}}+N_{\max\_\mathrm{slope}}\times T$$

As shown in Figure 5, we compared fitting of measured values to the determined model and evaluated the accuracy of the model by calculating $A_{f}$ and $B_{f}$ between the observed and predicted values (Table 4). The developed model predicted the change of measured acidity well by using the same model constants and reflected the seasonal initial acidity value ($N_{0}$).

#### 3.3.2. Dynamic Model and Validation

To predict the acidity change of Kimchi under fluctuating temperature conditions, Equations (2)–(4) and (7) were adapted by defining $N_{\max}$ as the $N_{\max}$ value of the temperature at each time. The prediction results were compared to the acidity measurement results under fluctuating temperature conditions that were not used for model development (24 h intervals for 20 days at 0–10 °C and 14 days at 5–15 °C), as shown in Figure 6. The accuracy of the model was evaluated using $A_{f}$ and $B_{f}$ (Table 5). $A_{f}$ and $B_{f}$ were close to 1 under both fluctuating temperature conditions, confirming the accuracy of the model. However, while the developed model is suitable for evaluating the acidity of Kimchi below the maximum acidity, the maximum value predicted by the model tends to decrease as the temperature decreases after about 15 and 8 days under fluctuating temperature conditions of 0–10 and 5–15 °C, respectively. To analyze this tendency, we carried out an additional temperature change experiment to verify the change in acidity in the stationary phase. Comparing the pH value of Kimchi under fluctuating temperature conditions of 5–15 °C with the acidity value under fixed temperature conditions for up to 30 days (Figure 7), it was confirmed that the maximum acidity value does not change even when the temperature changes in the stationary phase. Therefore, the assumption that $N_{\max}$ changes according to instantaneous food temperature is invalid.

In many foods, the maximum quality index in the stationary phase is constant regardless of temperature conditions. For example, in the case of pasteurized milk, it has been experimentally confirmed that the maximum microbial concentration in the stationary phase is constant regardless of the distribution temperature [33]. Therefore, the food quality can be predicted for a given time through the abovementioned stages. However, for Kimchi, the acidity changes according to the fermentation stage, and the microorganisms that grow during distribution have variable maximum microbial concentrations in the stationary phase depending on the distribution temperature history [16]. When the distribution temperature is constant, it is easy to predict the change in quality index by fixing the maximum quality index in the stationary phase. On the other hand, in fluctuating temperature environments, the maximum quality index changes depending on temperature. It is therefore, necessary to determine the maximum quality index in the stationary phase real-time according to the temperature history, thus, ensuring that the quality index can be determined using the existing food quality model.

#### 3.3.3. Dynamic Model Using MKT

We compared the measured temperature of Kimchi using a data logger with the estimated MKT history (see Figure 8). It was confirmed that variation of MKT decreases gradually over time, even in the case of variations at the 10 °C actual temperature. It then maintains a constant value with time when acidity reaches the stationary phase. This implies that MKT does not change much even if the temperature changes in the stationary phase; hence, the maximum acidity change is relatively constant.

Figure 9 compares the predicted values using the quality model based on the assumption that $N_{\max}=N_{\max}\left(T_{\mathrm{MKT}}\right)$ to Kimchi acidity measurement data under fluctuating temperature conditions of 0–10 and 5–15 °C. In contrast to the case using the assumption $N_{\max}=N_{\max}\left(T\right)$, the acidity value of the Kimchi predicted by the model did not decrease as the temperature decreased in the stationary phase. Moreover, the acidity changes of Kimchi were well predicted. The accuracy of the model was calculated using $A_{f}$ and $B_{f}\;$for observed and predicted acidity results using the $N_{\max}=N_{\max}\left(T_{\mathrm{MKT}}\right)$ assumption (Table 6). The values were closer to 1 than under the $N_{\max}=N_{\max}\left(T\right)$ assumption, demonstrating the improved accuracy.

## 4. Conclusions

We developed a method to predict Kimchi ripening according to temperature to provide information on how the ripening changes during distribution. The physicochemical changes were analyzed according to temperature, and acidity (%) was selected as an appropriate freshness index. Acidity (%) is dependent on temperature and correlates well with the sensory data. Moreover, it is easy to measures and reproducible in the field. A predictive model was developed using the Baranyi and Roberts and Polynomial models to mathematically predict the change of acidity (the selected quality index) over time. When the developed model was applied with fluctuating temperature conditions, the acidity value of the Kimchi predicted by the model decreases as the temperature changes from high to low. To solve this problem, a method using a mean kinetic temperature (MKT) is proposed. The results show that the accuracy of the model is high, since $A_{f}\;\mathrm{and}\;B_{f}$, which show the similarity between the measured and predicted values, are close to 1. It was confirmed that there is no great variation in the maximum acidity change when using MKT, since the MKT does not change much if the temperature changes in the stationary phase where the maximum acidity is constant. This study provides important information on the development of models to predict changes in food quality index under fluctuating temperature conditions.

The developed kinetic model uniquely treated the quality index at the stationary phase to be a function of MKT. It could successfully reproduce the observations under both constant and varying temperature conditions. The suppliers could decide on the sales and disposal of Kimchi in circulation based on the model prediction using the monitored temperature histories for their products. Both suppliers and consumers could prepare the temperature histories to make Kimchi of their favorite flavor. The developed model can be applied to other products such as beef for which the quality index at the stationary phase changes with temperature histories. Further investigations for the possible applications of the model on other foods and verification are necessary. At the moment, beef also has similar temperature-dependent stationary phase levels problems [16,17].

## Figures and Tables

**Figure 1 foods-09-01075-f001:**
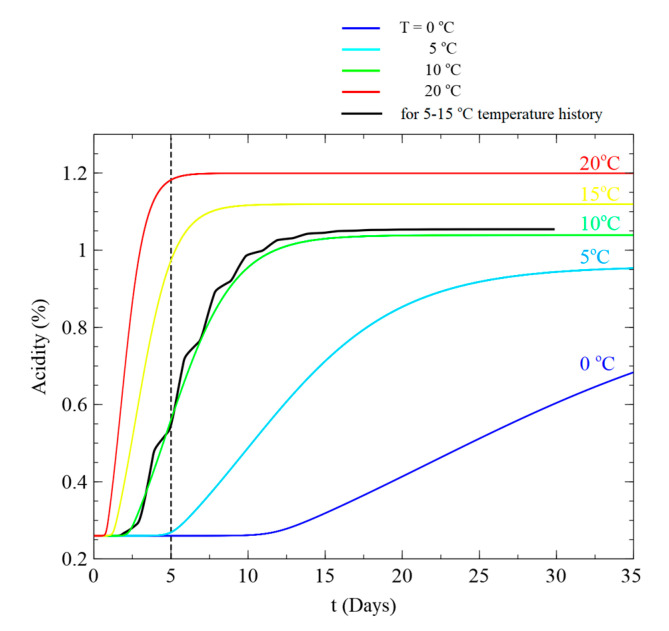
Example of estimation of mean kinetic temperature.

**Figure 2 foods-09-01075-f002:**
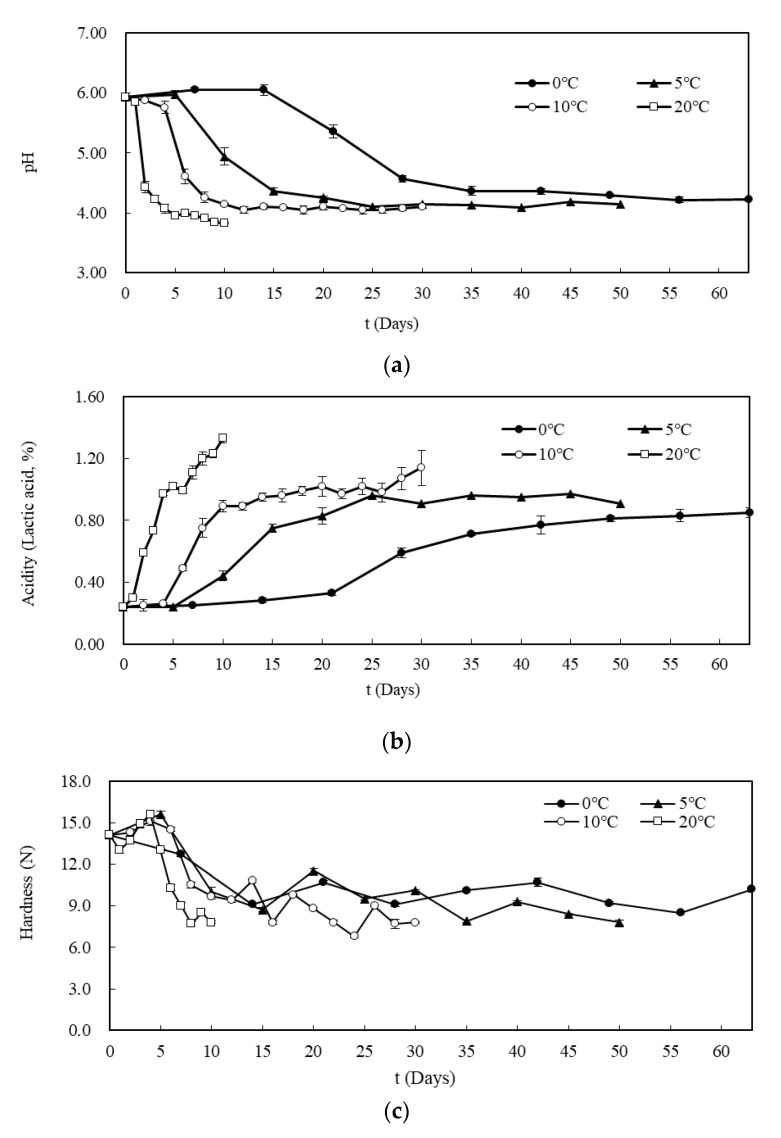
Changes in (**a**) pH, (**b**) acidity, and (**c**) hardness of Kimchi during fermentation at 0, 5, 10 and 20 °C.

**Figure 3 foods-09-01075-f003:**
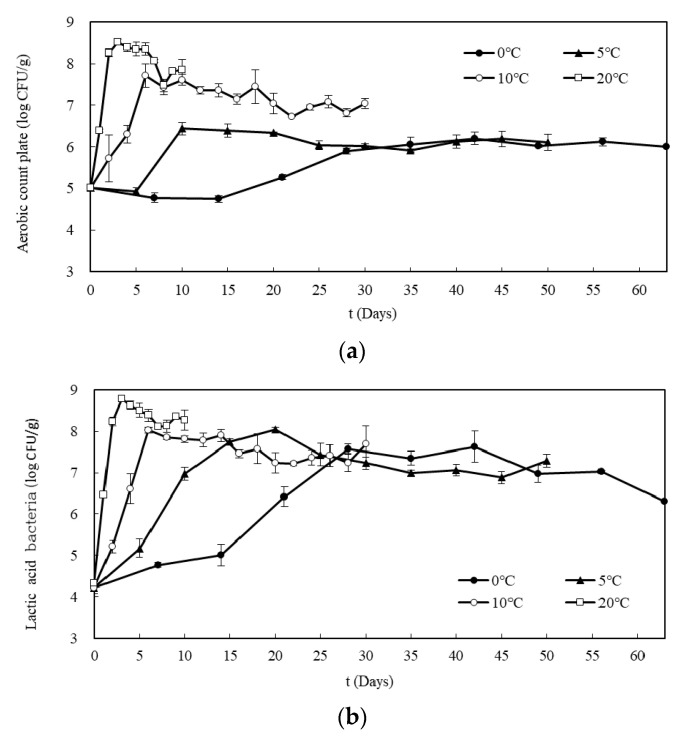
Changes in (**a**) aerobic count plate and (**b**) lactic acid bacteria of Kimchi during fermentation at 0, 5, 10, and 20 °C.

**Figure 4 foods-09-01075-f004:**
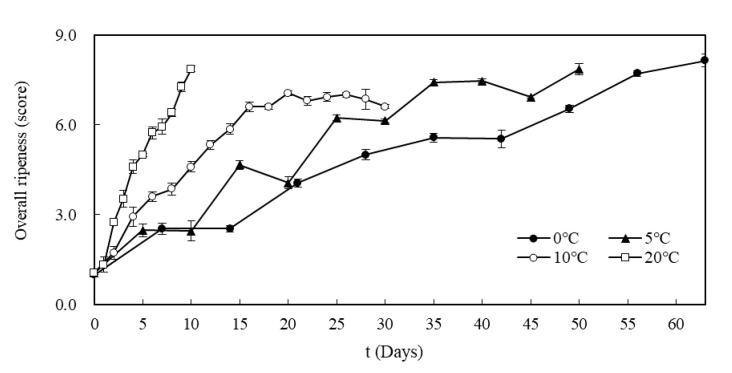
Comparison of the overall ripeness of Kimchi change over time at 0, 5, 10, and 20 °C.

**Figure 5 foods-09-01075-f005:**
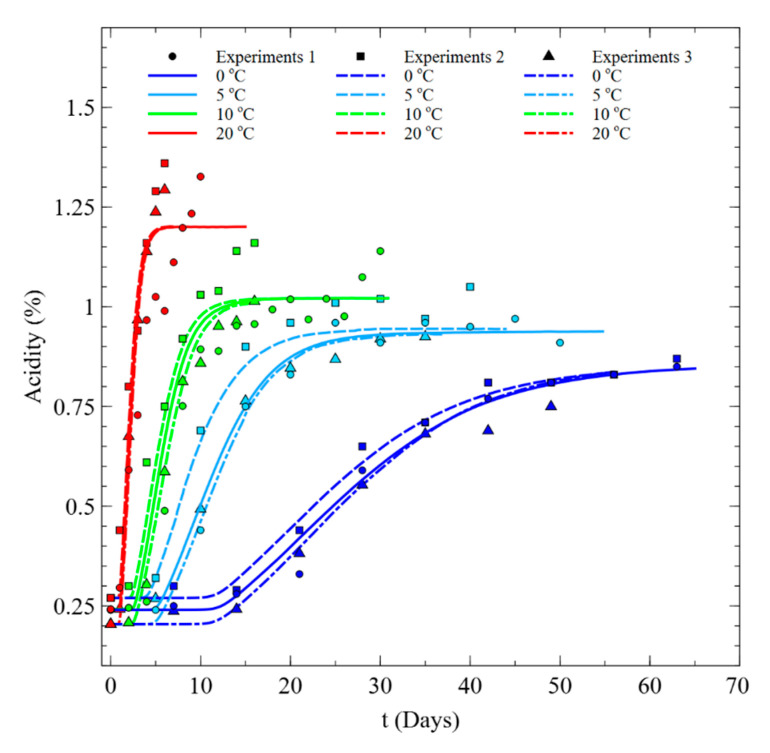
Measured and fitted acidity variation of Kimchi with time under different fixed temperature conditions. The symbols show the measured data and the lines stand for the fitted data. The different symbols, i.e., circles, squares, and triangles, and different line styles, i.e., solid, dashed, dashed dot, are used to represent data from different batch of experiments. The different colors categorize different constant temperature conditions.

**Figure 6 foods-09-01075-f006:**
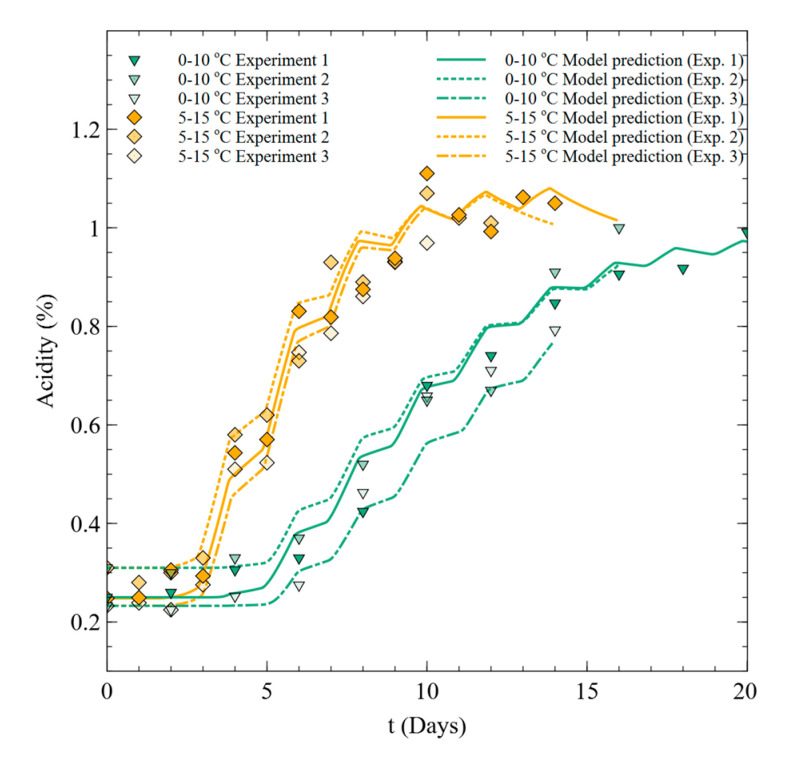
Comparison of measured acidity variation and predictions of the model with the assumption of $N_{\max}=N_{\max}\left(T\right)$ for experiments 1–3.

**Figure 7 foods-09-01075-f007:**
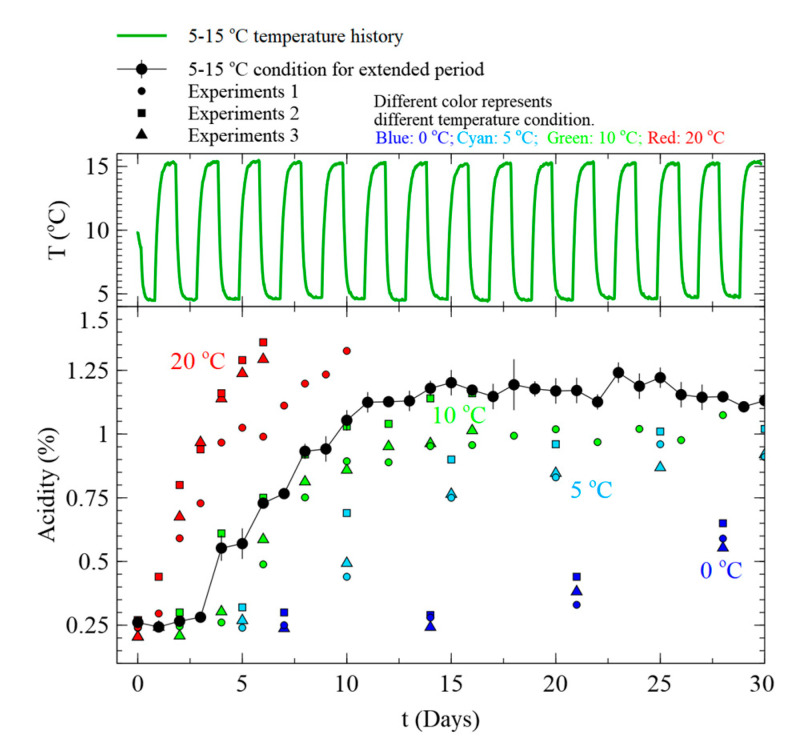
Variation of $N_{\max}$ under fluctuating temperature conditions. The black circles represent the measured acidity variation under fluctuating temperature conditions. The green line shows the fluctuating temperature history tested. The black circles with connecting lines and error bars represent the observed acidity variation with time and the measured standard deviation at each point. The different symbols, i.e., circles, squares, and triangles, are used to represent data from different batch of experiments. The different colors categorize different constant temperature conditions.

**Figure 8 foods-09-01075-f008:**
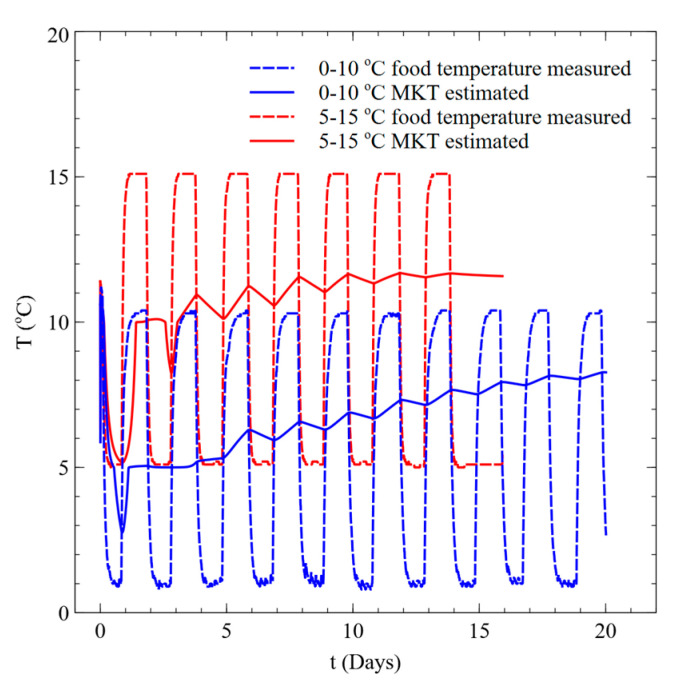
Measured temperature vs. estimated mean kinetic temperature.

**Figure 9 foods-09-01075-f009:**
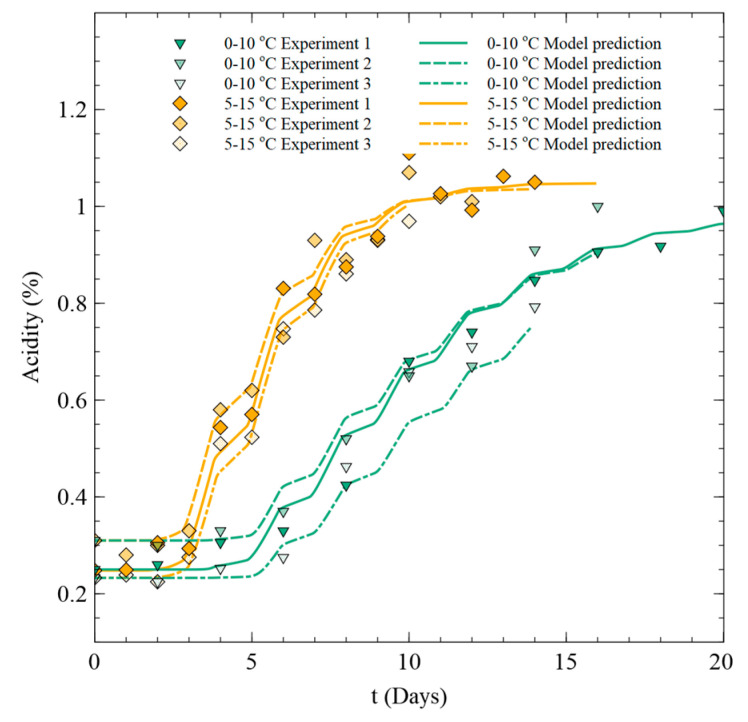
Comparison of measured acidity variation and predictions of the model with the assumption of $N_{\max}=N_{\max}\left(T_{MKT}\right)$ for experiments 1–3.

**Table 1 foods-09-01075-t001:** Changes in sensory evaluation of Kimchi during fermentation at 0, 5, 10 and 20 °C.

Temperature (°C)	Fermentation Period (Day)	Sensory Evaluation				
		Appearance (Softness)	Sour Smell	Sour Taste	Crunchiness	Overall Ripeness
0	0	1.00f	1.06e	1.20e	8.86a	1.00f
	7	2.53e	2.46de	2.60d	7.46b	2.53e
	14	3.06e	2.66d	2.53d	6.86bc	2.53e
	21	4.33d	3.53d	3.40d	6.00cd	4.06d
	28	5.00d	4.60bc	4.80c	5.73de	5.00cd
	35	6.00c	4.21c	6.21b	5.21def	5.57bc
	42	6.46bc	5.00bc	6.40b	5.06defg	5.53bc
	49	7.06ab	4.66bc	6.73b	4.26fg	6.53b
	56	7.85a	6.00ab	7.92a	4.78efg	7.71a
	63	8.06a	6.78a	8.14a	4.00g	8.14a
5	0	1.00f	1.06d	1.20f	8.86a	1.06e
	5	2.80e	2.26d	2.46e	7.73b	2.46d
	10	2.66e	2.06d	2.53e	7.46b	2.46d
	15	6.13c	4.26bc	6.33c	6.13c	4.66c
	20	4.73d	4.06c	4.86d	6.40c	4.06c
	25	6.73bc	5.40abc	6.80bc	5.00d	6.20b
	30	7.40ab	5.13abc	7.40ab	5.00d	6.13b
	35	7.92a	5.85ab	8.14a	4.00de	7.42ab
	40	8.06a	6.53a	7.86a	4.13de	7.46ab
	45	7.46ab	5.66abc	7.66ab	4.53de	6.93bc
	50	8.20a	6.60a	8.20a	3.73e	7.86a
10	0	1.00f	1.06f	1.20g	8.86a	1.06g
	2	1.53f	1.46ef	2.00g	8.06ab	1.73g
	4	3.26e	2.73de	3.13f	7.33bc	2.93f
	6	5.00d	2.93de	4.80e	6.46cd	3.60f
	8	5.13d	3.06cd	5.20e	6.40cd	3.86ef
	10	5.66cd	4.00bcd	5.66de	6.03de	4.60de
	12	6.60bc	4.53abc	6.53cd	5.53def	5.33cd
	14	6.73abc	5.06ab	7.00bc	5.33efg	5.86bc
	16	7.13ab	5.20ab	7.66ab	4.73fg	6.60ab
	18	6.73abc	5.00ab	6.66bc	5.06efg	6.60ab
	20	7.40ab	5.26ab	7.46abc	4.80fg	7.06a
	22	7.66ab	5.26ab	7.40abc	4.40g	6.80ab
	24	7.86a	5.80a	8.20a	4.80fg	6.93a
	26	7.20ab	5.86a	7.46abc	4.66fg	7.00a
	28	7.26ab	5.00ab	7.60abc	4.80fg	6.86ab
	30	7.20ab	5.13ab	7.53abc	5.20efg	6.60ab
20	0	1.00e	1.06d	1.13f	8.86a	1.06g
	1	1.33e	1.06d	1.20f	8.80a	1.33g
	2	3.53d	2.60cd	4.26e	7.73a	2.73f
	3	5.00c	3.26bc	5.26de	6.53b	3.53ef
	4	5.86bc	4.60ab	6.00cd	6.26b	4.60de
	5	6.66ab	4.86ab	7.20ab	5.93b	5.00d
	6	6.53ab	4.66ab	6.86bc	5.33bc	5.73cd
	7	7.06ab	4.66ab	7.46ab	5.93b	5.93cd
	8	7.20ab	5.26a	7.60ab	5.33bc	6.40bc
	9	7.73a	5.73a	7.93ab	4.46c	7.26ab
	10	7.93a	5.86a	8.40a	4.53c	7.86a

Different letters within the same column differ significantly (*p* < 0.05).

**Table 2 foods-09-01075-t002:** Pearson correlation coefficients between overall ripeness and quality index of Kimchi.

Quality Index	Storage Temperature (°C)	Correlation Coefficient (r)	Statistical Significance ^†^
pH	0	−0.920	**
	5	−0.887	**
	10	−0.884	**
	20	−0.862	**
Total acidity (% lactic acid)	0	0.915	**
	5	0.932	**
	10	0.947	**
	20	0.989	**
Texture (hardness)	0	0.308	NS
	5	0.150	NS
	10	0.147	NS
	20	0.105	NS
Hunter *L* value	0	−0.111	NS
	5	−0.045	NS
	10	0.026	NS
	20	0.128	NS
Hunter *a* value	0	0.807	**
	5	0.532	**
	10	0.536	**
	20	0.603	**
Hunter *b* value	0	0.559	NS
	5	−0.096	NS
	10	−0.066	NS
	20	−0.013	NS
Aerobic count plate (Log CFU/g)	0	0.892	**
	5	0.505	NS
	10	0.595	*
	20	0.572	NS
Lactic acid bacteria (Log CFU/g)	0	0.860	**
	5	0.628	*
	10	0.674	**
	20	0.665	*

^†^ Significance: ** *p* < 0.01; * *p* < 0.05; NS—not significant.

**Table 3 foods-09-01075-t003:** Model constants for fitted model and their 95% confidence interval.

Model Parameters	Fitted Value	Confidence Interval	
		2.5%	97.5%
$a_{0}$	7.09$\;\times 10^{-2}$	6.00$\;\times 10^{-2}$	8.38$\;\times 10^{-2}$
$a_{1}$	1.52$\;\times 10^{-2}$	9.86$\;\times 10^{-3}$	2.34$\;\times 10^{-2}$
$a_{2}$	2.33$\;\times 10^{-3}$	1.90$\;\times 10^{-3}$	2.87$\;\times 10^{-3}$
$Q_{0}$	3.70$\;\times 10^{-7}$	1.70$\;\times 10^{-8}$	8.08$\;\times 10^{-6}$
$N_{\max\_\mathrm{slope}}$	1.60 $\times \;10^{-2}$	1.28$\;\times 10^{-2}$	2.02$\;\times 10^{-2}$
$N_{\max\_\mathrm{intercept}}$	8.78$\;\times 10^{-1}$	8.33$\;\times 10^{-1}$	9.27$\;\times 10^{-1}$

**Table 4 foods-09-01075-t004:** Comparison of accuracy and bias factors of the predictive models from experiments 1 to 3.

Experiment	Temperature Condition (°C)	*A_f_* ^†^	*B_f_* ^‡^
1	0	1.04	1.01
	5	1.04	1.00
	10	1.05	1.01
	20	1.04	1.01
2	0	1.05	0.95
	5	1.06	0.97
	10	1.10	1.07
	20	1.09	0.92
3	0	1.04	1.03
	5	1.12	1.06
	10	1.13	0.91
	20	1.06	0.95

*A_f_*^†^—accuracy factor; *B_f_*^‡^—bias factor.

**Table 5 foods-09-01075-t005:** Comparison of accuracy and bias factors of the predictive models using the assumption $N_{\max}=N_{\max}\left(T\right)$ for experiments 1 to 3.

Experiment	Temperature Condition (°C)	*A_f_* ^†^	*B_f_* ^‡^
1	0–10	1.07	1.03
	5–15	1.08	1.06
2	0–10	1.07	0.97
	5–15	1.05	0.99
3	0–10	1.06	1.04
	5–15	1.05	1.01

*A_f_*^†^—accuracy factor; *B_f_*^‡^—bias factor.

**Table 6 foods-09-01075-t006:** Comparison of accuracy and bias factors of the predictive models using the assumption of $N_{\max}=N_{\max}\left(T_{\mathrm{MKT}}\right)$ for experiments 1–3.

Experiment	Temperature Condition (°C)	*A_f_* ^†^	*B_f_* ^‡^
1	0–10	1.07	1.02
	5–15	1.07	1.04
2	0–10	1.08	0.96
	5–15	1.05	0.98
3	0–10	1.05	1.03
	5–15	1.04	1.00

*A_f_*^†^—accuracy factor; *B_f_*^‡^—bias factor.

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
