# Peer review of "Development of Dynamic Model for Real-Time Monitoring of Ripening Changes of Kimchi during Distribution"

_foods, 2020, doi:10.3390/foods9081075_

Round 1

Reviewer 1 Report

The manuscript presents the results of an interesting study related to the prediction of changes of food quality index under fluctuating temperature conditions. The manuscript is very robust, showing both quality measurements and modeling, however, in my opinion not all the figures are necessary.

The data of the experiment 1,2,3 if I understand well, are the repetition of the same analysis, therefore should be present on the same graphs (e.g. as in Figure 8), it’s not necessary to prepare a separate figure for each repetition.

Why didn’t you use exactly the same temperatures for storage of the Kimchi and for the modeling (1,5,10 and 20°C)?

How did you decide the end of storage for Kimchi at each temperature?

For texture analysis you should specify which test did you used, I guess penetration, but this information should be added. Moreover, the firmness (force) should be expressed in N not in kg.

Pag 6, line 206-I guess it should be 0°C not 5°C.

Pag 8, line 255, you stated that there were no significant differences in color parameter b, but you didn’t shown the ANOVA analysis. The same for other parameters.

Author Response

First of all, the authors appreciate all the comments from the reviewers, and find out that the comments are very helpful and crucial to better present the contents in the manuscript. The manuscript has been updated by faithfully reflecting all the comments. The questions from reviewers and the responses are shown in black and red respectively in this response letter and they are shown in red in the revised manuscript. The responses to the individual comment from both the first and second reviewers are listed as following:

Responses to the comments of the reviewer 1

The manuscript presents the results of an interesting study related to the prediction of changes of food quality index under fluctuating temperature conditions. The manuscript is very robust, showing both quality measurements and modeling, however, in my opinion not all the figures are necessary.

(Responses) Thank you for the positive evaluation. According to your comments the number of figures was decreased by merging Figs. 5 and 6(a)-(c) in the original manuscript into Fig. 4 in the revised one. Figures 7(a)-(c) and 10(a)-(c) in the old manuscript are combined into Figures 5 and 8, respectively in the revised manuscript.

Figure 4. Measured and fitted acidity variation of Kimchi with time under different fixed temperature conditions.

Figure 5. Comparison of measured acidity variation and predictions of the model with the assumption of  for experiments 1-3.

Figure 8. Comparison of measured acidity variation and predictions of the model with the assumption of  for experiments 1-3.

The data of the experiment 1,2,3 if I understand well, are the repetition of the same analysis, therefore should be present on the same graphs (e.g. as in Figure 8), it’s not necessary to prepare a separate figure for each repetition.

(Responses) Thank you for the suggestion. The separate in Figs. 7 and 10 are merged into Figs. 6 and 9, respectively. Please refer to the revised manuscript and the responses to the previous comments.

Why didn’t you use exactly the same temperatures for storage of the Kimchi and for the modeling (1,5,10 and 20°C)?

(Responses) Thank you for the comments. They were the same conditions for the storage and for the modeling. The nominal temperatures were used for the storage temperature, where the mean temperatures were used for the modeling. To avoid the confusion, the nominal temperatures only were used in the revised manuscript. Please refer to the updated manuscript and the figures therein.

How did you decide the end of storage for Kimchi at each temperature?

(Responses) The following paragraph was added in the revised manuscript.

According to the Codex Alimentarius, a collection of international food standards, the composition of acidity for fermented kimchi should not be more than 1 %. Accordingly, the end of storage for Kimchi was generally selected to be the time for the acidity to approach the value of 1%. However, the observation period was extended to consider the plateaus of the stationary phase in this study. Kimchi generates carbon dioxide in the process of fermentation, and the polyethylene-made wrap could burst. These were considered in choosing the end of storage.

For texture analysis you should specify which test did you used, I guess penetration, but this information should be added. Moreover, the firmness (force) should be expressed in N not in kg.

(Responses) Thank you for the comments. The unit of the hardness was changed to N and the related numbers were converted accordingly. Below is the updated Fig. 2(c).

(c)

Figure 2. (c) hardness of Kimchi during fermentation at 0, 5, 10 and 20 °C.

Pag 6, line 206-I guess it should be 0°C not 5°C.

(Responses) Thank you for the comments. The manuscript was updated accordingly.

Pag 8, line 255, you stated that there were no significant differences in color parameter b, but you didn’t shown the ANOVA analysis. The same for other parameters.

(Responses) The table below represents the result of the ANOVA analysis and the letters following the numbers represent the group categorized by the statistical analysis. For example, the “L” value at 0 °C were all in group a, which means that the “L” value could not differentiate the change of the food quality varying with the storage time. It is the same for other Hunter color values “a” and “b” since they were divided into only two groups and they cannot differentiate the food quality variation with precision.

However, other reviewer (reviewer #3) suggested that it would be better to make the manuscript compact for the better presentation of the materials. So, the table is not included in the revised manuscript now. If you consider that the inclusion of all ANOVA results is necessary, then the tables will be added to the manuscript. Instead, the following sentence was added to the manuscript.

The ANOVA analysis of the data revealed that the Hunter color values were not an adequate quality index to differentiate the food quality change over time with precision. The details of the analysis and graph related to Hunter color values were not included for the sake of presentation’s clearness.  

Table. Changes in Hunter color of Kimchi during fermentation at 0, 5, 10 and 20 °C

Temperature (°C)

Fermentation period

(day)

Hunter color value

L

a

b

0

0

32.52a

14.78a

15.81a

7

31.84a

18.09b

17.51b

14

31.60a

18.59b

17.56b

21

32.55a

17.99b

17.65b

28

31.67a

19.03b

17.47b

35

31.87a

18.53b

17.55b

42

33.30a

19.50b

17.97b

49

31.21a

19.11b

17.13b

56

32.05a

20.02b

17.70b

63

32.09a

18.18b

17.38b

5

0

32.52a

14.78a

15.81a

5

31.54a

17.70ab

17.04ab

10

31.44a

17.91ab

17.37ab

15

31.50a

18.07ab

17.04ab

20

32.76a

18.57b

17.85b

25

32.83a

20.35b

18.23b

30

32.69a

18.98b

17.53ab

35

32.58a

48.96b

17.59b

40

31.79a

18.69b

17.21ab

45

31.66a

20.72b

17.97b

50

31.17a

19.86b

17.38ab

10

0

32.52a

14.78a

15.81ab

2

28.07b

16.10ab

15.53a

4

32.24b

18.12abc

17.50bc

6

33.83b

18.35abc

18.60c

8

31.98b

18.66abc

17.76c

10

32.27b

18.89abc

17.67c

12

31.61b

18.15abc

17.13abc

14

32.10b

18.72abc

17.77c

16

31.73b

20.08bc

17.80c

18

32.41b

19.76bc

17.89c

20

31.47b

20.47c

17.67c

22

32.56b

20.25c

18.04c

24

31.57b

20.42c

17.82c

26

31.64b

18.71abc

17.71c

28

31.94b

19.28bc

17.68c

30

31.29b

21.04c

17.66c

20

0

32.52a

14.78a

15.81a

1

32.55a

14.42a

15.12a

2

33.35a

19.70b

17.80b

3

33.65a

20.23b

18.96b

4

32.91a

21.12bc

18.63b

5

32.31a

20.06b

18.23b

6

32.07a

21.06bc

18.38b

7

33.23a

20.34b

18.66b

8

33.01a

21.31bc

18.74b

9

34.27a

22.35bc

19.18b

10

33.62a

21.28bc

18.93b

Different letters within the same column differ significantly (p < 0.05).

Reviewer 2 Report

  1. Elements of scientific novelty should be presented in a detailed and convincing manner (in the last paragraph of the Introduction, and shortly in Abstract).
  2. Potential application of developed method should be mentioned in Introduction or Conclusion.
  3. Little bits of minor English corrections  required through the manuscript.
  4. Conclusion is fine, but not much in the way of future recommendations, thus, I recommend to extend this section.

Author Response

First of all, the authors appreciate all the comments from the reviewers, and find out that the comments are very helpful and crucial to better present the contents in the manuscript. The manuscript has been updated by faithfully reflecting all the comments. The questions from reviewers and the responses are shown in black and red respectively in this response letter and they are shown in red in the revised manuscript. The responses to the individual comment from both the first and second reviewers are listed as following:

Responses to the comments of the reviewer 2

  1. Elements of scientific novelty should be presented in a detailed and convincing manner (in the last paragraph of the Introduction, and shortly in Abstract).

(Responses) Thank you for the comments. The scientific novelty of the developed model and its potential applications are stated in the “Introduction,” “Abstract,” and “Conclusions” sections. Please refer to the revised manuscript.

The following paragraph wad added in the Abstract.

The developed kinetic model uniquely treated the quality index at the stationary phase as a function of MKT. The predictions using the food temperature histories could help suppliers and consumers make a reasonable decision on the sales, storage, and consumption of foods. The developed model could be applied to other products such as beef for which the quality index at the stationary phase also changes with temperature histories.

The introduction section was revised and the following paragraph was added.

The developed kinetic model uniquely treated the quality index at the stationary phase to be a function of MKT. It could successfully reproduce the observations under both constant and varying temperature conditions. This will help the suppliers decide on the sales and disposal of Kimchi in circulation based on the model prediction using the monitored temperature histories for their products. Both suppliers and consumers could prepare the temperature histories to make Kimchi of their favorite flavor. The developed model can also be applied to other products such as beef for which the quality index at the stationary phase changes with temperature histories.

The conclusions section was revised and the following paragraph was added.

The developed kinetic model uniquely treated the quality index at the stationary phase to be a function of MKT. It could successfully reproduce the observations under both constant and varying temperature conditions. The suppliers could decide on the sales and disposal of Kimchi in circulation based on the model prediction using the monitored temperature histories for their products. Both suppliers and consumers could prepare the temperature histories to make Kimchi of their favorite flavor. The developed model can be applied to other products such as beef for which the quality index at the stationary phase changes with temperature histories. Further investigations for the possible applications of the model on other foods and verification are necessary. At the moment, beef also have a similar temperature-dependent stationary phase levels problems [16,17].  

  1. Potential application of developed method should be mentioned in Introduction or Conclusion.

(Responses) Thank you for the comments. The scientific novelty of the developed model and its potential applications are stated in the “Introduction,” “Abstract,” and “Conclusions” sections. Please refer to the revised manuscript.

The following paragraph wad added in the Abstract.

The developed kinetic model uniquely treated the quality index at the stationary phase as a function of MKT. The predictions using the food temperature histories could help suppliers and consumers make a reasonable decision on the sales, storage, and consumption of foods. The developed model could be applied to other products such as beef for which the quality index at the stationary phase also changes with temperature histories.

The introduction section was revised and the following paragraph was added.

The developed kinetic model uniquely treated the quality index at the stationary phase to be a function of MKT. It could successfully reproduce the observations under both constant and varying temperature conditions. This will help the suppliers decide on the sales and disposal of Kimchi in circulation based on the model prediction using the monitored temperature histories for their products. Both suppliers and consumers could prepare the temperature histories to make Kimchi of their favorite flavor. The developed model can also be applied to other products such as beef for which the quality index at the stationary phase changes with temperature histories.

The conclusions section was revised and the following paragraph was added.

The developed kinetic model uniquely treated the quality index at the stationary phase to be a function of MKT. It could successfully reproduce the observations under both constant and varying temperature conditions. The suppliers could decide on the sales and disposal of Kimchi in circulation based on the model prediction using the monitored temperature histories for their products. Both suppliers and consumers could prepare the temperature histories to make Kimchi of their favorite flavor. The developed model can be applied to other products such as beef for which the quality index at the stationary phase changes with temperature histories. Further investigations for the possible applications of the model on other foods and verification are necessary. At the moment, beef also have a similar temperature-dependent stationary phase levels problems [16,17].

  1. Little bits of minor English corrections required through the manuscript.

(Responses) The manuscript was closely checked and corrected. Please refer to the revised manuscript.

  1. Conclusion is fine, but not much in the way of future recommendations, thus, I recommend to extend this section.

(Responses) Thank you for the comments. The scientific novelty of the developed model and its potential applications are stated in the “Introduction,” “Abstract,” and “Conclusions” sections. Please refer to the revised manuscript.

The following paragraph wad added in the Abstract.

The developed kinetic model uniquely treated the quality index at the stationary phase as a function of MKT. The predictions using the food temperature histories could help suppliers and consumers make a reasonable decision on the sales, storage, and consumption of foods. The developed model could be applied to other products such as beef for which the quality index at the stationary phase also changes with temperature histories.

The introduction section was revised and the following paragraph was added.

The developed kinetic model uniquely treated the quality index at the stationary phase to be a function of MKT. It could successfully reproduce the observations under both constant and varying temperature conditions. This will help the suppliers decide on the sales and disposal of Kimchi in circulation based on the model prediction using the monitored temperature histories for their products. Both suppliers and consumers could prepare the temperature histories to make Kimchi of their favorite flavor. The developed model can also be applied to other products such as beef for which the quality index at the stationary phase changes with temperature histories.

The conclusions section was revised and the following paragraph was added.

The developed kinetic model uniquely treated the quality index at the stationary phase to be a function of MKT. It could successfully reproduce the observations under both constant and varying temperature conditions. The suppliers could decide on the sales and disposal of Kimchi in circulation based on the model prediction using the monitored temperature histories for their products. Both suppliers and consumers could prepare the temperature histories to make Kimchi of their favorite flavor. The developed model can be applied to other products such as beef for which the quality index at the stationary phase changes with temperature histories. Further investigations for the possible applications of the model on other foods and verification are necessary. At the moment, beef also have a similar temperature-dependent stationary phase levels problems [16,17].

Reviewer 3 Report

General comments:

This manuscript is no doubt a very thorough description of the development of models to predict changes in the food quality of Kimchi during changing temperature conditions in the period of time from production until it reaches the consumer. This is interesting and valuable information for producers of Kimchi, and similar products as well as distributors and others who handle Kimchi in the market place. 

The English language has barely any mistakes, and is well written. 

That said, this is a very detailed and compiled study, it may be easy to get a little lost. One reason, I believe, was that some information is missing, mislabelled, or not labelled clearly. I was not able to trace back to the experiments 1 to 3, and what they were described as. This made it more challenging to make sense of the figures including these experiments. 

As the material is large, perhaps it would be an idea to consider removing one or two of the figures? For example, Figure 3 a,b,c, could be sufficient as text only. 

On the other hand, for the sensory results, it might be interesting to present the Overall ripeness in a line plot, with days on the X-axis, and the 9-point scale on the Y-axis, to compare the changes due to different storage temperatures.

For sensory analysis in general, as for descriptions of other instrumental analyses, it needs to be presented with appropriate details, for example, the number of hours of training, number of samples per session, how many sensory replicates, serving order design, in what kind of setting was the test being performed, was rinsing agents served in between samples, etc. This will make the presentation of the sensory results more reliable.

In more details: 

Abstract: 

Line 13 - Change .. Kimchi is changes to changing

1. Introduction:

Line 53 - Suggest change to Kim et al., (2019)

Line 64 - Suggest change to McMeekin et al., 2013

2. Materials and methods

Line 130 - Change sensual to sensory 

Line 131 - Are there any definitions connected to each of these attributes?

3. Results

Line 286 - Did you mean softness or tenderness?

Author Response

First of all, the authors appreciate all the comments from the reviewers, and find out that the comments are very helpful and crucial to better present the contents in the manuscript. The manuscript has been updated by faithfully reflecting all the comments. The questions from reviewers and the responses are shown in black and red respectively in this response letter and they are shown in red in the revised manuscript. The responses to the individual comment from both the first and second reviewers are listed as following:

Responses to the comments of the reviewer 3

This manuscript is no doubt a very thorough description of the development of models to predict changes in the food quality of Kimchi during changing temperature conditions in the period of time from production until it reaches the consumer. This is interesting and valuable information for producers of Kimchi, and similar products as well as distributors and others who handle Kimchi in the market place.

The English language has barely any mistakes, and is well written.

(Responses) Thank you for the positive evaluation.

That said, this is a very detailed and compiled study, it may be easy to get a little lost. One reason, I believe, was that some information is missing, mislabelled, or not labelled clearly. I was not able to trace back to the experiments 1 to 3, and what they were described as. This made it more challenging to make sense of the figures including these experiments.

(Responses) Thank you for the comments. To clearly state the experiment labels, the sentence in the original manuscript was rewritten as:

To reflect seasonal changes, Kimchi was purchased and tested in spring (March to May), summer (August to September), and winter (November to January), and the tests were labeled as experiments 1, 2 and 3, respectively.

As the material is large, perhaps it would be an idea to consider removing one or two of the figures? For example, Figure 3 a,b,c, could be sufficient as text only.

(Responses) Thank you for the comments. Figure 3 was removed, and the contents became more compact and eassier to read.

On the other hand, for the sensory results, it might be interesting to present the Overall ripeness in a line plot, with days on the X-axis, and the 9-point scale on the Y-axis, to compare the changes due to different storage temperatures.

(Responses) The requested figure of the overall kimchi ripeness change was added.

Figure 4. Changes in overall ripeness of Kimchi during fermentation at 0, 5, 10 and 20 °C.

For sensory analysis in general, as for descriptions of other instrumental analyses, it needs to be presented with appropriate details, for example, the number of hours of training, number of samples per session, how many sensory replicates, serving order design, in what kind of setting was the test being performed, was rinsing agents served in between samples, etc. This will make the presentation of the sensory results more reliable.

(Responses) The section on sensory analysis was completely rewritten. Please refer to the following paragraph as well as the revised manuscript.

The sensory evaluation of Kimchi was carried out using a 9-point scale for 20 trained sensual assessors. In this case, the trained participants are selected from staff and researchers (5 males, 15 females, aged 30-60) at the Korea Food Research Institute. Beforehand, the participants were trained to differentiate kimchi qualities based on temperature and evaluate the appearance (softness), sour smell, sour taste, crunchiness, and overall ripeness of the Kimchi. For instance, "appearance" pertains to the color shade-softness of the Kimchi where "sour smell" described the sensation of typical generated flavor. The taste was used to obtain desired sour effects, and the ripeness was used to draw an overall freshness of the product. The intensity of each item was ranked from 1 (very low) to 9 (very strong); 3, 5, and 7 points were awarded for low, normal, and high intensities, respectively. The 3-gram samples were provided in white polyethylene cups and numbered with three-digit randomly. Each sample was served one by one at room temperature.

In more details:

Abstract:

Line 13 - Change .. Kimchi is changes to changing

(Responses) Updated accordingly.

  1. Introduction:

Line 53 - Suggest change to Kim et al., (2019)

(Responses) The references and their list are updated according to the journal standards. Please refer to the updated reference style and their list in the revised manuscript.

Line 64 - Suggest change to McMeekin et al., 2013

(Responses) The references and their list are updated according to the journal standards. Please refer to the updated reference style and their list in the revised manuscript.

  1. Materials and methods

Line 130 - Change sensual to sensory

(Responses) Updated accordingly.

Line 131 - Are there any definitions connected to each of these attributes?

(Responses) The section on sensory analysis was completely rewritten. Please refer to the following paragraph as well as the revised manuscript.

The sensory evaluation of Kimchi was carried out using a 9-point scale for 20 trained sensual assessors. In this case, the trained participants are selected from staff and researchers (5 males, 15 females, aged 30-60) at the Korea Food Research Institute. Beforehand, the participants were trained to differentiate kimchi qualities based on temperature and evaluate the appearance (softness), sour smell, sour taste, crunchiness, and overall ripeness of the Kimchi. For instance, "appearance" pertains to the color shade-softness of the Kimchi where "sour smell" described the sensation of typical generated flavor. The taste was used to obtain desired sour effects, and the ripeness was used to draw an overall freshness of the product. The intensity of each item was ranked from 1 (very low) to 9 (very strong); 3, 5, and 7 points were awarded for low, normal, and high intensities, respectively. The 3-gram samples were provided in white polyethylene cups and numbered with three-digit randomly. Each sample was served one by one at room temperature.

  1. Results

Line 286 - Did you mean softness or tenderness?

(Response) Softness is the better fit for the quality index of Kimchi. The manuscript was updated accordingly.